# Neuro-Urology and Biobanking: An Integrated Approach for Advancing Research and Improving Patient Care

**DOI:** 10.3390/ijms241814281

**Published:** 2023-09-19

**Authors:** Sander M. Botter, Thomas M. Kessler

**Affiliations:** 1Swiss Center for Musculoskeletal Biobanking, Balgrist Campus AG, 8008 Zürich, Switzerland; 2Department of Neuro-Urology, Balgrist University Hospital, University of Zürich, 8008 Zürich, Switzerland; tkessler@gmx.ch

**Keywords:** neuro-urology, neurogenic lower urinary tract dysfunction, lower urinary tract symptoms, biobank, biomarker, biorepository, biospecimen, translational research

## Abstract

Understanding the molecular mechanisms underlying neuro-urological disorders is crucial for the development of targeted therapeutic interventions. Through the establishment of comprehensive biobanks, researchers can collect and store various biological specimens, including urine, blood, tissue, and DNA samples, to study these mechanisms. In the context of neuro-urology, biobanking facilitates the identification of genetic variations, epigenetic modifications, and gene expression patterns associated with neurogenic lower urinary tract dysfunction. These conditions often present as symptoms of neurological diseases such as Alzheimer’s disease, multiple sclerosis, Parkinson’s disease, spinal cord injury, and many others. Biobanking of tissue specimens from such patients is essential to understand why these diseases cause the respective symptoms and what can be done to alleviate them. The utilization of high-throughput technologies, such as next-generation sequencing and gene expression profiling, enables researchers to explore the molecular landscape of these conditions in an unprecedented manner. The development of specific and reliable biomarkers resulting from these efforts may help in early detection, accurate diagnosis, and effective monitoring of neuro-urological conditions, leading to improved patient care and management. Furthermore, these biomarkers could potentially facilitate the monitoring of novel therapies currently under investigation in neuro-urological clinical trials. This comprehensive review explores the synergistic integration of neuro-urology and biobanking, with particular emphasis on the translation of biobanking approaches in molecular research in neuro-urology. We discuss the advantages of biobanking in neuro-urological studies, the types of specimens collected and their applications in translational research. Furthermore, we highlight the importance of standardization and quality assurance when collecting samples and discuss challenges that may compromise sample quality and impose limitations on their subsequent utilization. Finally, we give recommendations for sampling in multicenter studies, examine sustainability issues associated with biobanking, and provide future directions for this dynamic field.

## 1. Introduction

### 1.1. Overview of Neuro-Urology and Its Importance in Healthcare

Neurogenic lower urinary tract dysfunction is highly prevalent, affects the life of millions of people worldwide, and causes considerable economic burden [1,2]. Neuro-urology, a highly specialized field at the intersection of neurology and urology, plays a crucial role in understanding and managing neurogenic lower urinary tract dysfunction (NLUTD) [3]. The coordination between the central and peripheral nervous system across parasympathetic, sympathetic, and somatic pathways is vital for maintaining normal urinary function. Disruptions in this delicate balance leads to a wide range of NLUTD, necessitating specialized evaluation and management.

One of the key aspects of neuro-urology is the comprehensive assessment of patients with NLUTD. This involves a thorough medical history, physical examination, and diagnostic testing to identify the underlying cause of urinary tract dysfunction. The evaluation may include urodynamic studies, which assess bladder function and the coordination between the bladder and the urethra during the filling and emptying phases [4,5]. Neuroimaging techniques, such as magnetic resonance imaging (MRI) [6] or computed tomography (CT) [7], may also be utilized to visualize the structures or the control of the urinary tract and identify anatomical or neurological abnormalities.

The findings from these assessments guide the development of tailored treatment plans for patients with neuro-urological conditions. The management strategies employed in neuro-urology aim to preserve upper urinary tract function, to improve quality of life, to control urinary tract infections, and to maintain a low-pressure reservoir that is both continent and capable of emptying completely [1,2,3].

Neuro-urology’s importance in healthcare extends beyond the realm of diagnosis and treatment. It also plays a crucial role in advancing our understanding of the mechanisms underlying urinary tract disorders. By studying the complex interactions governing bladder filling and emptying, involving the nervous system, urinary tract anatomy, and musculature in the lower urinary tract, researchers can gain valuable insights into the pathophysiology of these conditions, leading to the development of novel diagnostic techniques [8] and innovative therapies [9,10,11]. In recent years, large efforts have been made by the International Continence Society (ICS) to standardize treatment guidelines of NLUTD [12,13]. However, current treatment strategies differ between centers [14] and many guidance gaps fail to address key issues in NLUTD care, including possibilities for cathether reuse in low-resourced countries, comorbid conditions frequently associated with NLUTD, such as sexual and bowel dysfunction, care transitions from pediatric to adult urological clinics, and surveillance protocols for NLUTD patients, leaving many without appropriate monitoring in real-world settings [15]. Hence, these research programmes contribute to the development of evidence-based guidelines and new standards of patient care. In many such studies, biomarker discovery is an important tool to reach this goal.

### 1.2. Introduction to Biobanking and Its Relevance to Neuro-Urological Research

Biobanking involves systematically and reproducibly collecting, processing, storing, and managing biological samples, such as tissues, blood, urine, and DNA, along with associated clinical and demographic data. These samples and data are preserved in biorepositories, known as biobanks, for use in medical research. Biobanks provide access to well-characterized and consented samples representing NLUTD and various diseases with neuro-urological complications, including Alzheimer’s disease, multiple sclerosis, and spinal cord injury. Tissue banks play a crucial role in advancing scientific knowledge in these areas, as they form the key to discovering biomarkers, which may help to better understand disease mechanisms, to develop into new diagnostic approaches, or to evaluate the effectiveness of different interventions [16,17].

With respect to neuro-urology research, implementing biobanking as part of a research study offers numerous advantages. Firstly, the establishment of large-scale, longitudinal collections of biological specimens from patients with different types and stages of urinary disorders allows further disease discrimination and an understanding of temporal differences in affected tissues such as the bladder wall. In addition, through cross-sectional comparisons of samples from affected individuals and healthy controls, researchers can highlight which differences may be associated with disease susceptibility and disease prognosis. Finally, the tracking of these differences in clinical trials testing new treatment intervention schemes allows to see the effect of treatment at the molecular level and can ultimately assist in the development of novel therapeutic strategies.

When considering the number of publications within the last 10 years of neuro-urological research (excluding malignant cancer) in which biobanking was part of the study design, we found a total of 1605 publications within the last 10 years (see Appendix A). By exporting these abstracts and analyzing their key words, the main topics in these studies became apparent (see Figure 1A). When stratifying for different disease types, biobanking efforts in neuro-urological studies were mostly represented in spinal cord injury studies (45%), followed by studies involving patients with spina bifida (24%) or with diabetic neuropathy (12%) (Figure 1B).

It is important to realize that the “biobanking” and “biobanks” are both terms with no legal status, and that there is no agreement on a precise definition of biobanking. This essentially means that any collection of human biological materials may be called a biobank [18]. For professional biobanks, it is nowadays essential to possess a minimum technical standard and comprehensive protocols that address various aspects related to sample collection, processing, storage, and distribution. These protocols should encompass laboratory safety, quality management, data protection and data access, governance, obtaining informed consent, and many other relevant considerations [19]. Also nowadays, high-quality biobanking can be ensured through national certification efforts (e.g., certifications provided by the Swiss Biobanking Platform initiative; www.swissbiobanking.ch, accessed on 10 August 2023) or by achieving accreditation to international standards, such as ISO 20387:2018 [20] (also see Section 3.2). Importantly, best practice biobanking requires adherence to ethical guidelines, following informed consent procedures, upholding privacy and data protection regulations, and maintaining secure and proper sample storage and management practices [21]. Establishing standardized protocols and promoting collaboration between biobanks and researchers, such that sample collections collected at different institutions may be combined, are vital not only for maximizing the value of biobanking in medical research, but also for ensuring a return of interest on the initial public costs involved in establishing the sample collections. 

## 2. Applications of Biobanking in Neuro-Urology

### 2.1. Specimen Types Collected and Recommended Collection Procedures

Neuro-urology biobanks gather a diverse array of specimens to conduct comprehensive studies on urinary disorders. These specimens include urine, blood, and tissue samples, such as bladder biopsies. Of these, urine specimens offer non-invasive access to biomarkers, metabolites, and cellular components that reflect the physiological and pathological status of the urinary system [22,23]. Blood plasma or serum, derived by centrifugation from whole blood samples, provide valuable information about systemic factors and circulating biomarkers relevant to neuro-urological conditions. By isolating specific blood cells, the patients’ immunological system can be studied. Tissue samples, obtained through surgical procedures or biopsies, allow for detailed histopathological and molecular analyses. Genetic material, including DNA and RNA isolated from both blood and tissue samples aids in identifying genetic predispositions and studying gene expression profiles. Collectively, these specimens provide a comprehensive resource for researchers to explore the underlying mechanisms of urinary disorders.

Because in many cases, the extent of effects of a given intervention are unknown, some studies undertake a holistic approach. This means that not only urine, but also blood, stool, bladder biopsies [24], and depending on the patient cohort, cerebrospinal fluid samples [25], are considered in the study design.

One specific aim of collecting biospecimens is the search for biomarkers that can reflect current disease state and disease progression, and may be used as a tool, in addition to careful clinical examination, to monitor treatment response [26,27,28,29]. Some often used biomarkers in neuro-urological research and their recommended collection procedures are described below.

#### 2.1.1. Neurofilaments

Neurofilaments are proteins found in nerve cells and their levels can be measured in cerebrospinal fluid (CSF) or blood. Elevated levels of neurofilaments have been associated with neuronal damage or degeneration in various neurological conditions, including those affecting the urogenital system [30]. Monitoring neurofilament levels can provide insights into the extent of neurological damage and give insight into renal functioning in these disease cohorts [31].

The collection of CSF requires a lumbar puncture procedure performed by a trained healthcare professional and should follow standard aseptic techniques to minimize the risk of contamination [32,33]. It is crucial to follow proper collection procedures to minimize the risk of hemolysis or contamination, as these factors can affect the accuracy of the measurements. Samples are usually collected in sterile tubes or containers and immediately processed according to the specific assay requirements. This usually includes a centrifugation and aliquotation step, after which samples are stored at −80 °C until further use.

Blood samples for neurofilament analysis can be collected using standard venipuncture techniques into appropriate blood serum or plasma collection tubes. In case blood serum tubes are used, clotting agents within the tubes can considerably speed up the clotting process, but nevertheless care must be taken to allow sufficient clotting time according to the manufacturer’s specifications. If blood plasma tubes containing anticoagulants like citrate or EDTA are utilized, samples should be processed promptly after collection. Typically, a single centrifugation step is adequate for separating the serum or plasma from its cellular components. The resulting serum or plasma is then divided into aliquots and stored at −80 °C. It is recommended for studies to consistently use the same tube type, either blood serum or blood plasma, across their research period and various sites as this will minimize technical variation, although combined analysis of neurofilament measurements in plasma and serum samples is possible [34].

#### 2.1.2. Cytokines and Inflammatory Markers

Inflammatory markers, such as cytokines, can be measured in blood or urine samples. Inflammatory processes often accompany neurological conditions and urogenital disorders. Monitoring the levels of specific cytokines or markers of inflammation can help assess the presence and severity of inflammation and guide treatment strategies [35]. Cytokines and inflammatory markers can be measured in blood or urine samples (e.g., [36]). Urine samples for cytokine and inflammatory marker analysis are typically collected using clean catch midstream urine collection methods to minimize contamination. Specialized urine collection containers or tubes may be used. Urine samples should be placed on ice immediately after collection. Depending on the study aim, the samples can be directly divided into aliquots and stored at −80 °C. Alternatively, e.g., in case the urine-containing microbiome should be concentrated, the samples are first centrifuged prior to separate aliquotation and cryostorage of the supernatant and sediment-containing pellet.

#### 2.1.3. Specific Urinary Tract Biomarkers

Whilst the previously mentioned biomarkers are often found elevated as part of systemic (inflammatory) or central (Alzheimer) diseases leading in addition to urinary tract problems, several specific urinary tract biomarkers produced within the urinary tract itself can provide insights into disease progression. For example, elevated levels of the protein Neural Growth Factor (NGF), produced by the bladder urothelium and bladder smooth muscle cells, may indicate renal dysfunction or bladder pathology [37]. Other examples include urinary markers related to smooth muscle stretch, such as urinary ATP, or specific metabolites, such as the purine derivative hypoxantin [38]. Many more biomarkers have meanwhile been described. It is not the aim of this review to provide a comprehensive overview of all currently studied biomarkers, however, the authors refer to some excellent recently published overviews covering this topic (e.g., [29,35,39]). For all sample collection and aliquot procedures described above, due care must be taken to safely perform sample processing steps, preferably making use of a sterile biosafety level two work bench, and to use proper labeling and storage techniques, preferably using preprinted, barcoded aliquot tubes and tube scanners.

### 2.2. Applications in Multicenter Clinical Trials

Multicenter studies in neuro-urological research offer valuable opportunities to study larger patient populations, increase statistical power, and enhance the generalizability of research findings. Our center is currently involved in several multicenter clinical trials in which tissue samples are being banked. These studies are aimed at preventing neurogenic detrusor overactivity and detrusor sphincter dyssynergia emergence and consequent damage to the upper urinary tract in spinal cord injury patients through stimulation of the tibial nerve (TASCI, (https://clinicaltrials.gov/, accessed on 10 August 2023) NCT03965299 [24,40]), as well as several initiatives to overcome urinary tract infections using non-antibiotic treatment strategies (CAUTIphage, ImmunoPhage, mTORUS: [41,42,43,44,45]. These multicenter studies pose significant challenges, described in more detail below, in several aspects pertaining to collection, storage, and distribution of collected tissue samples. Addressing these challenges requires careful planning, standardized protocols, effective communication, and collaborative efforts among researchers, clinicians, and study coordinators across participating centers.

#### 2.2.1. Heterogeneity within Patient Populations

Patients may be biologically heterogeneous in terms of symptoms, or by having different etiologies leading to similar phenotypes that are then diagnosed as belonging to the same disease, but clinical heterogeneity is also an important factor, as non-standardized clinical input and outcome measures create technical variation that obscures disease progression of treatment response, making tracking of these processes difficult. For instance, with respect to urinary tract infections (UTI), there has been a lack of standardization in the definition of UTIs in neuro-urological patients [46]. This lack of standardization is also found in other patient cohorts [47,48]. This heterogeneity makes it difficult to standardize patient populations across multiple research centers, leading to variability in data collection and introducing confounding factors, all potentially affecting the validity and generalizability of study findings. Achieving protocol standardization in neuro-urological research is challenging due to variations in clinical practices, diagnostic criteria, and treatment approaches among different centers. This problem can be solved by creating unified guidelines [49] and standardizing treatment protocols, as well as through extensive disease group stratification strategies prior to clinical trial enrollment, involving deep clinical phenotyping, imaging, and biobanking [50].

#### 2.2.2. Site Selection and Recruitment

Identifying and selecting appropriate research centers that have the necessary expertise and resources to participate in the study can be challenging. When undertaking biobanking efforts, each study center must in addition have certain infrastructure considerations that need to align with the study’s objectives; for instance, regular maintenance of devices such as centrifuges and freezers must take place, and backup strategies in case of power failure must be in place. Prior to commencing multicenter studies involving biobanking, it is highly recommendable to organize site visits to create an overview of local infrastructure and to gain insight in local logistic processes, such as the distance and time needed between taking samples from a patient in the operating theatre and/or hospital ward and freezer. It is equally important to make sure that all sample processing is performed according to a centralized Standard Operating Procedure (SOP) and that processing efforts of the different sites are harmonized prior to the trial commencing, e.g., by organizing parallel sample processing or sample analysis with a single parental bio-sample as the source [51]. Finally, it is recommended to utilize a centralized data entry system capable of receiving (patient-sensitive) encrypted sample data from all locations, such as REDCap (Research Electronic Data Capture) [52]. Alternatively, if technical or ethical limitations prevent the implementation of a central system, it is crucial to establish a consensus on the procedures for data entry, data harmonization, and sharing among the sites.

#### 2.2.3. Regulatory and Ethical Considerations

Research protocols, informed consent processes, and data protection procedures may vary between different regions or countries. Ensuring compliance with regulatory requirements and ethical standards across multiple centers and jurisdictions is a time-consuming challenge, and is unsurprisingly associated with increased costs [53]. This issue is not only related to the neuro-urological research field and several initiatives aimed at streamlining this process have been proposed within other research areas [54,55,56,57].

#### 2.2.4. Logistics

If the study involves the collection of biological samples or specimens, logistics related to their collection, storage, transportation, and analysis can be complex, and the potential effects of logistics on sample quality are often underestimated. For instance, the distance between the operating theatre or nursing facilities and the laboratory may influence sample processing times (timepoint sample taken from patient until timepoint storage of sample in a −80 °C freezer) and hence, sample quality may differ between centers depending on the biomarker of interest. To optimize logistics processes, effective communication among all stakeholders is vital. Also, the study center overseeing a clinical trial in which biobanking is performed should be aware of potential differences or bottlenecks between centers with respect to logistical processes and propose alternative proceedings if necessary. Logistical challenges may also include scheduling site visits, coordinating data collection activities, or coordinating sample shipments between centers. Coordinating timelines and maintaining effective communication among multiple centers is vital for maintaining sample integrity and a smooth execution of the study.

### 2.3. Translation into Clinical Applications

Findings from neuro-urology biobanking studies can have a significant impact on clinical practice and contribute to the development of personalized treatment strategies and targeted interventions. There are several ways in which these findings may be translated into clinical applications:**Understanding molecular mechanisms**: Biobanking has played a pivotal role in elucidating the molecular mechanisms underlying urinary disorders. By analyzing the collected specimens, researchers can investigate genetic variations, gene expression patterns, epigenetic modifications, and protein profiles associated with specific neuro-urological conditions, such as in Alzheimer’s disease or bladder pain syndrome [31,58,59,60]. These investigations contribute to a deeper understanding of disease pathophysiology and potential therapeutic targets.**Biomarker-guided diagnosis and prognosis:** Neuro-urology biobanking studies can identify biomarkers associated with specific neuro-urological conditions analogous to other urological conditions, e.g., acute kidney injury [61]. These biomarkers can aid in the early diagnosis, classification, and prognosis of patients [62,63]. Clinicians can then utilize these biomarkers to improve diagnostic accuracy, predict disease progression, and adjust treatment plans accordingly.**Identification of drug targets and development of novel therapies**: By studying human and animal tissue samples, researchers can identify novel drug targets, validate existing targets, and develop new therapies that specifically address the underlying mechanisms of neuro-urological conditions such as overactive bladder syndrome [64,65,66].**Personalized treatment and treatment tailoring approaches**: Neuro-urology biobanking studies contribute to the development of personalized medicine approaches [17]. By analyzing the genetic, molecular, and clinical data from biobanked samples, researchers can identify patient subgroups with distinct characteristics or treatment needs. This knowledge can guide the selection of appropriate therapies and dosage adjustments or the use of combination therapies tailored to individual patients whilst minimizing adverse effects.

## 3. Challenges and Considerations in Neuro-Urology Biobanking

### 3.1. Privacy and Data Protection

Biobanking raises important ethical and legal considerations regarding informed consent, privacy, and data protection [67]. It is crucial to ensure that appropriate ethical guidelines are followed and that consent is obtained from patients prior to sample collection and usage. As biobanking involves the storage and utilization of personal health information, strict protocols must be implemented to protect patient privacy and ensure compliance with data protection regulations. Anonymization and encryption techniques should be employed to safeguard sensitive information and prevent unauthorized access [68]. This is important as participants are more likely to join a study if biobanks can ensure that participants’ identities and sensitive information remain confidential and protected [69]. Challenges include obtaining a valid signed informed consent [70], implementing robust security measures to safeguard data from unauthorized access or breaches [71], and navigating legal and ethical regulations governing data privacy. At our institution, the latter aspect is managed by a specialized team of study coordinators. Their role involves facilitating communication among researchers, physicians, and the local ethical board. Moreover, these coordinators oversee the “encryption key” that links patients to their corresponding (sample) codes. This system guarantees the protection of personal information, shielding it from researchers who are engaged in working with these samples and for whom such information should not be disclosed.

### 3.2. Standardization and Quality Assurance

Maintaining high-quality standards in biobanking is essential to ensure the reliability and reproducibility of research findings. However, although biobanking efforts have become prevalent worldwide, there has been no standardization of storage practices or modes of operation. Given that the prevalence of irreproducible biomedical research is substantial [72,73], standardization, especially in multicenter studies, is a key aspect that biobanks should adhere to.

The lack of reproducibility is known to be due in part to differences in the integrity of biospecimens, caused by differences in collection procedures and preanalytical processing [74]. This in turn can directly affect biomarker sensitivity and/or specificity in clinical evaluations which undermines the effective integration of these biomarkers into clinical settings. In neuro-urological research, as in many other research fields, the transitioning into clinical practice of even the most promising biomarkers often remains unclear [29].

As the standardization of sample collection, processing, storage, and data-management protocols is crucial to minimize preanalytical variations, biobanks should adhere to standard procedures as much as possible. Preanalytical processing should include a detailed documentation of well-described standardized processing steps. For instance, in our institute, each single sample processing is accompanied by filling in a dedicated custom-made working instruction, outlining all the processing steps, the time stamps for collection and storage, temperatures, centrifuge speeds, and the number of aliquots derived from the parental sample. All processing steps need to be marked after completion and all deviations from the standard protocol are noted. Finally, all sample-specific working instructions are stored as separate files and are part of the metadata of each individual sample.

The principles of standardization and quality assurance can also be implemented within the biobanking infrastructure. Access to biobank storage facilities should ideally be restricted to relevant biobank staff. Staff members must be well trained for proper sample collection, registration, processing, and storage, and consistent evaluation of such training must be assessed and recorded. Additionally, a professional biobank information management system (BIMS) will ensure the accurate positioning, traceability, and recording of non-conformity incidents for each aliquot.

While urine seems to show limited effects from freeze-thaw cycles [75,76], it is generally important to minimize temperature fluctuations during storage, as they might adversely impact biomarker stability in other tissue types [77,78]. Therefore, a modern cryo-infrastructure is a fundamental necessity for the prolonged storage of samples. Mechanical freezers should be connected to emergency backup systems using liquid carbon dioxide or liquid nitrogen to reduce the risk of thawing in the event of freezer failure. Furthermore, ongoing monitoring of the sample storage temperatures and the temperature profiles of each freezer should be carried out to prevent any overlooked thawing events. Unnecessary freeze-thaw cycles can also be mitigated by segmenting samples into separate aliquots, preferably using cryotubes either equipped with specialized cryogenic storage labels or with a preprinted code, thus circumventing manual inscription and potential sample allocation errors.

Quality assurance measures, including regular monitoring of sample integrity, are imperative to maintain a high quality of stored specimens and the associated data. Nowadays, several accreditation and certification strategies for biobanks exist. Recommendations such as the NCI’s “Best Practices for Biospecimen Resources” [79] have since 2018 evolved in the ISO 20387 standard for biobanking [80,81]. ISO 20387 is an international standard that provides guidelines for the application of biometric technology in the context of testing laboratories and their operations. Published in 2018 by the International Organization for Standardization (ISO), the biobanking standard aims to ensure the accuracy, reliability, and integrity of biometric testing processes. It covers various aspects of biometric technology testing, including test methodologies, test environments, and test reporting. It also provides guidelines for performance metrics and emphasizes the need for proper management systems, competent personnel, the calibration and maintenance of equipment, and protecting the confidentiality and integrity of the test data and results.

Due to its relative newness and specificity, not all countries provide certification or accreditation processes for ISO 20387. Instead, alternative certification processes might be accessible; for instance, in Switzerland, the Swiss Biobanking Platform (www.swissbiobanking.ch, accessed on 10 August 2023) has introduced its own quality management certification labeling systems, “Norma” and “Optima,” which closely mirror the ISO 20387 standard prerequisites. In addition, international biobanking organizations such as Biological and Biomolecular Resources Research Infrastructures (BBMRI-ERIC; www.bbmri-eric.eu, accessed on 10 August 2023) stimulate standardization by offering quality management courses. Through upholding these infrastructural and procedural standards, biobanks not only guarantee high-quality sampling, but can showcase their professional work approach that may aid to establish trust among academic and commercial users.

### 3.3. Further Challenges in Sample Collection

Besides the challenges alread discussed in previous paragraphs, collecting tissues from neuro-urological cohorts can present many other challenges. To maximize the quality and utility of the collected samples, these must be considered in the study design or in conversations with the biobank performing the biosampling.
**Invasive procedures**: Collecting tissue samples from neuro-urological cohorts may require invasive procedures, e.g., the collection of cerebrospinal fluid or the collection of biopsies that can only be taken through surgery. These procedures carry inherent risks and may not be feasible for all patients, especially in case such surgeries need to be organized in addition to standard clinical procedures.**Competition between patient cohorts:** As the incidence of some patient cohorts in neuro-urological studies is relatively low, for instance in traumatic spinal cord injury [82], there may be competition for inclusion of the patients’ samples in several studies. In such cases, a feasibility analysis may be necessary prior to conducting the study.**Use of (co-)medication**: Patients with neuro-urological conditions frequently receive a range of specific medications to manage their condition and related symptoms. These treatments might encompass antimuscarinic drugs and alpha-blockers for addressing an overactive bladder and urinary retention, as well as intracavernous injections of alprostadil, papaverine, and phentolamine in cases of erectile dysfunction [2]. In addition, co-medication of more common drugs such as immunosuppressant or antidepressants may be prescribed. These drugs may introduce confounding factors and complicate the interpretation of research results, as they may impact the biochemical or molecular composition of tissues. For instance, lipid levels, evaluated in several neuro-urological patient cohorts [83,84,85], are known to be influenced by some common medications [86,87]. To mitigate the impact of co-medication, strategies such as stratifying samples based on medication profiles may be considered, analyzing the effects of specific medications separately, or accounting for medication use as a variable in statistical analyses. Importantly, the potential impact of (co-)medication should be a subject of conversation between researchers, the biobank, and clinicians before biosampling commences.

### 3.4. Cost Coverage of Biobanking

As may have become clear from the previous paragraphs, biobanking efforts are labor- and time intensive, and hence, are associated with costs. At the same time, the tasks and costs involved in biobanking are often overlooked by researchers. For this reason, the long-term economic sustainability of biobanks requires careful consideration.

Like most technology support platforms, biobanks rely on some form of cost coverage to remain operable. The reimbursement, whether in monetary terms or in kind, should account for the labor invested in sample processing and associated overhead costs, while maintaining the principle of refraining from pursuing profits due to ethical considerations. Expenses can be associated with the biobank’s structure, including upkeep, equipment replacement, and acquisition. Additionally, there are costs tied to sample processing, such as consumables, customs import fees, and shipping charges (e.g., dry ice and courier expenses), as well as many other tasks [88]. Finally, salary costs for personnel time invested in sample processing, aliquoting, storage, and shipment also constitutes a significant cost factor [89].

These expenses may often be perceived by independent investigators as “supplementary costs,” given that these responsibilities were previously undertaken by students or staff members who were already remunerated. In addition, many investigators still prefer to have full control over their specimens [90]. As a result, a biobank might face challenges in convincing researchers that the costs associated with biobanking should be recognized not only as an investment to ensure high-quality biobanking for the current project but also, when employing broad consent usage, for future endeavors. Additionally, this approach offers a time-efficient solution as the tasks associated with biobanking no longer require staff members’ involvement.

To cover these costs, contemporary biobanks adhere to a business model allowing them to decide which costs should be enumerated and to predict their long-term viability. Several of such models have been described, ranging from full cost coverage to only covering consumables and sample handling [16,88]. Preferably as early as when securing funds for a study, researchers should take into consideration the costs of biobanking for their studies. This in return means that, ahead of the actual sample collection, it becomes the biobank’s duty to present the researcher with a comprehensive outline of the anticipated costs, such that researchers can use this estimate in their project budget overview. To do this, the biobank needs to discuss with the researcher in detail the collection and processing procedures. Ideally, and particularly crucial in multicenter studies, preliminary testing is conducted on a limited number of samples. This test encompasses the complete biobank process, starting from sample collection and extending to analysis. This approach should reveal the complete scope of activities linked to the biobanking of the samples, including potential technical or logistical issues, but should also offer a clearer depiction of the corresponding costs.

If costs escalate beyond the initial project budget, the biobank must promptly notify the researcher and proceed with sample collection and processing only upon receiving approval for the updated budget overview. Such open and clear communication will discourage researchers from viewing biobanks as mysterious financial burdens. Instead, it will likely increase the likelihood that researchers contemplating the utilization of the biobank’s services for upcoming endeavors will use the biobank, rather than attempting to start from scratch. Given that many projects are funded by public resources, the ultimate result of these actions is a more efficient allocation of project funds and the maintenance of standardized sample processing procedures.

### 3.5. Environmental Impact of Biobanking

The global demand for energy is ever-increasing; the same holds for biobanks.

As more and more samples are being stored for extended periods of time, one important challenge is to optimize the high energy expenditures of biobanks, mostly relating to storage and transportation of samples, and their impact on the environment. By estimating the energy costs per analyte and the total amount of analytes in the US alone, Shirakashi et al. reported the total energy expenditure to be 131 gigawatt-hours of electricity or 30,523 tons of CO_2_ into the atmosphere [91]. The authors calculated that the annual production of liquid nitrogen needed for one biobank alone already contributes to an emission of 0.925 tons of CO_2_ into the atmosphere and that one −80 °C freezer can consume up to 12 kWh (the equivalent of about 1 Swiss household [92]) of electricity each day. Finally, in addition to the energy expenses related to sample storage, the energy required for shipping samples across the globe is a significant factor that should not be overlooked.

Although these energy expenses are obviously necessary to keep a biobank operational, further streamlining of internal biobank processes may help to reduce energy costs and its impact on the environment. These may include:-Efficient sample retrieval strategies, such as the use of automated systems to retrieve samples, reducing the need for extended manual searches in which freezers need to be opened and closed regularly within a short time span;-The use of energy-efficient equipment, such as investments in energy-efficient, better-insulated freezers and refridgerators;-Data Center Optimization, such as a connected Laboratory Information Management Systems (LIMS) between study sites which omits the need for intermittent shipments of collected samples to a central storage facility. Instead, the connected system provides an overview of all sites and the samples only need to be collected in a central location during the analysis phase.

These strategies not only conserve energy but also enhance the overall efficiency as well as reducing the financial costs of biobank operations.

## 4. Future Directions and Emerging Technologies

### 4.1. Advancements in Biobanking Techniques

Continuous advancements in biobanking techniques are expected to further enhance neuro-urology research. Improvements in sample collection methods, preservation techniques, and sample processing automation can optimize the efficiency and quality of biobanking operations. New cryopreservation technologies, such as vitrification, in which elevated cryoprotectant concentrations and rapid cooling rates quickly convert a solution into a transparent non-crystalline solid compound [93,94] may be employed and more automation of sample processing will lead to less technical variation [95].

The integration of omics technologies, including genomics, transcriptomics, proteomics, and metabolomics, also holds tremendous potential in neuro-urology biobanking. Integrating multi-omics data can provide a holistic understanding of disease mechanisms, identify novel biomarkers, and predict disease progression [96]. These techniques are prone to soon be combined with new developments, such as artificial intelligence and machine learning algorithms [97], and assist in data analysis in large-scale datasets, such as pattern recognition of detrusor overactivity [98].

### 4.2. Advancements in Obtaining Consent

After a project has ended, more than once surplus sample collections have been left, and in case specific consent was applied, these samples may not be used anymore for other research purposes. This poses a problem as considerable effort and cost were invested in gathering and storing these samples. While broad consent permits secondary sample use in other contexts, participants typically favor specific consent over it [99]. Furthermore, participants usually interact solely with their study physician, remaining unaware of the biobank where their samples reside. This lack of transparency complicates requests for consent changes. A shift from local paper-based static consent to dynamic consent administered electronically (eConsent) is a solution. Dynamic consent empowers participants to provide, update, or withdraw consent autonomously and electronically [100]. It allows participants ongoing control over their data and participation, and enables real-time updates, informed decisions, and adjustments, promoting autonomy and transparent collaboration between researchers and participants.

Although paper-based tools remain relevant and will likely persist in the coming decades, digital health is swiftly progressing across healthcare sectors and will likely become the new standard for future generations. This encompasses electronic check-ins at medical facilities, automated data capture during clinical procedures, and centralized electronic patient records, such as made possible in Switzerland as of April 2017 (www.patientrecord.ch, accessed on 10 August 2023). This advancement enables comprehensive electronic interactions within healthcare contexts and will lower the threshold for adopting new systems such as eConsent [101]. Although this will likely create new challenges for biobanks as a withdrawn consent will mean that future use of samples is prohibited, the system allows the patient more control over its own biological material and is more transparent. Change rates of dynamic consents have been reported to be low [102]. More importantly, participants highly appreciated the opportunity to modify their informed consent preferences and it is believed that without this alternative, they would not have participated at all.

## 5. Conclusions

The utilization of biobanking in neuro-urology presents a unique opportunity to unravel the molecular mechanisms underlying urinary disorders. The systematic collection, storage, and analysis of biological specimens have facilitated a deeper understanding of disease pathogenesis, biomarker discovery, and personalized treatment approaches. However, ethical considerations, standardization, and data protection remain critical challenges that require careful attention. The future of neuro-urology biobanking holds immense potential, with advancements in biobanking techniques, advanced standardization efforts through accrediation or certification, and by embracing new ethical concepts empowering participants on their right to consent to their participation. Biobanking is a worthwhile endeavor for any research field. Those patients’ demographics and disease states that are represented in biobanking now will stand to benefit the most from future scientific discovery. By harnessing these advancements and fostering collaborative efforts between study centers, neuro-urology biobanking will assist to advance research and improve patient outcomes.

## Figures and Tables

**Figure 1 ijms-24-14281-f001:**
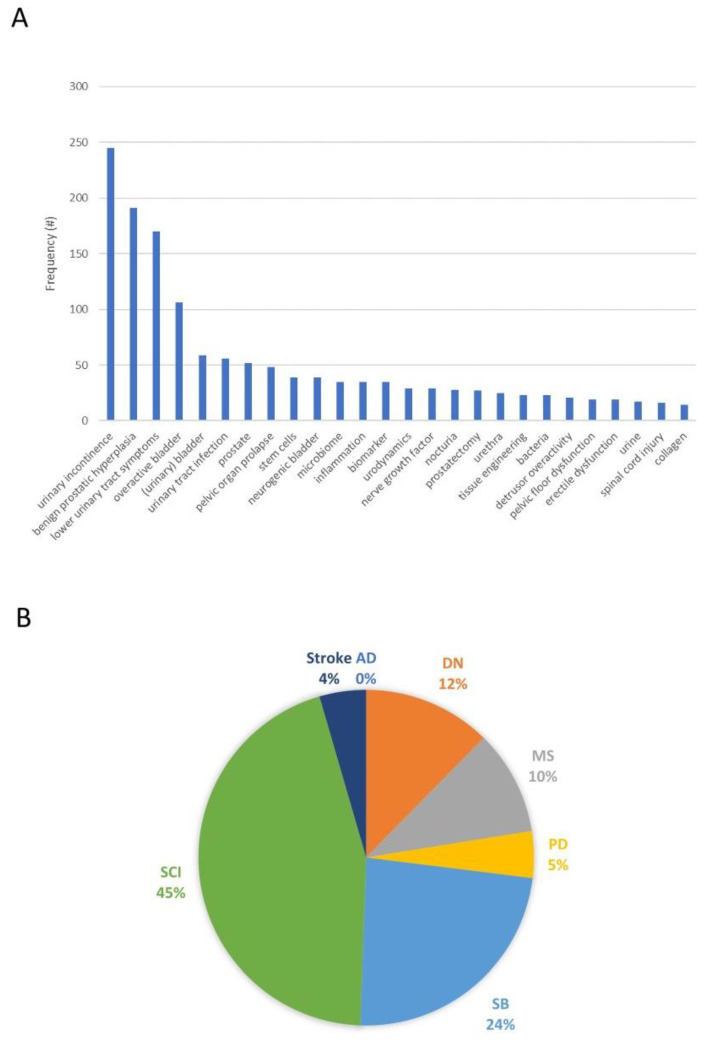
(**A**) Frequency distribution of 1400 keywords found in 1605 abstracts, selected through a systemic search in (Pubmed.gov, accessed on 10 August 2023) in search of all publications between 2012 and 2022 involving both biobanking/biomarkers and neuro-urological conditions (excluding malignant cancer). Using Pubmed’s export function of search results, 4816 keywords located within the 1605 abstracts were extracted and groups of keywords with a frequency >10 were summed. (**B**) The search strategy was extended to only include the following specific diseases relevant for the neuro-urological research field: Alzheimer’s disease (AD; 0 publications), diabetic neuropathy (DN; 11 publications), multiple sclerosis (MS; 9 publications), Parkinson’s disease (PD; 4 publications), spina bifida (SB; 21 publications), spinal cord injury (SCI; 40 publications), or stroke (4 publications). The full systematic search strategy has been described as Appendix A.

## Data Availability

All data reported in Figure 1 were derived from www.pubmed.gov, accessed on 10 August 2023.

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
