# Peer review of "Neuro-Urology and Biobanking: An Integrated Approach for Advancing Research and Improving Patient Care"

_ijms, 2023, doi:10.3390/ijms241814281_

Round 1

Reviewer 1 Report

The Authors review the relationship between biobanking and neurourology.  In particular they discuss the advantages and the role of biobanking for neurological studies and the types of specimen collected.  they underline particularly the legal challenges  of biobanking such as privacy and data protection, informed consent and eConsent, as well as the importance of standardization  and quality assurance when collecting samples,    challenges and possible bias  in multicenter studies and how to avoid them.  

The search of biomarkers in neurourology is a relatively new field, with a  growing interest over the past 10 years as documented in the recent literature.

In the field, this review is  new as its examines  the issue of biobanking from a pragmatic and pratical approach, mostly focused on the organizative, legal and technical aspects. 

However:

-          The Author  as in line 22-27 of the abstract,  write as  the use of biomarkers in the clinical pratice was already a reality, which is not yet true according to the majority of the literature [36]. Limits of current knowledge and limitation on the use of biobanking  for clinical pratice should be mentioned.

-          The Author  claim to discuss the application of  biobanking in traslational research.  The paragraph on “traslation into clinical practice” (line 302 -330 )  could be improved.  The sub-paragraph  “understanding molecular mechanism” conceptually should come before  “biomarker guided-diagnosis..”. Furthermore diseases in which biomarkers are shown to be potentially useful are always generally defined as “neuro-urological condition”. It is true that in  line 216 the Authors write “; it is not the aim of this review to provide a comprehensive overview of all currently studied biomarkers” however  it could be worth  mentioning the specific diseases in which biomarkers and biobanking  have proven  so far to be usefull.

Line 64  : modify from “bladder emptying” into  “bladder filing and emptying” - also because most studies on biomarkers in neurourology have focused on markers of OAB, which is a disorder of the filling phase

Line 67: the following two sentences deny the huge efforts made in recent years by national and International Scientific Societies and mostly by ICS  in order to standardize definition of NLUTD, and  diagnostic and therapeutical clinical pathways.  A phrase should be added

Line 76 to 87  and 88 to 97  : some concepts are repeated and the reading  is not fluid

Line 309 verify that reference 55 is correct

Line 311 verify that references 56-57 are correct

Line 269: SOP  the abbreviation is not explicated

Reviewer 2 Report

This is an excellent review of a complex and timely topic. I felt it was comprehensive and the methods were robust. 

Recommendations

- in line 68, you mention gaps in guidance. Please mention at some point in the paper what exactly guidance gaps need to be addressed

- please make a comment about the environmental sustainability issues related to biobanking

- please comment on the necessity of feasibility analysis especially in the setting of competing studies for rare diseases

The English in this study was easy to read with few errors. 

- in lines 181 and 182 I believe you mean clotting instead of clothing
